# A Virtual Reality System for Improved Image-Based Planning of Complex Cardiac Procedures

**DOI:** 10.3390/jimaging7080151

**Published:** 2021-08-19

**Authors:** Shujie Deng, Gavin Wheeler, Nicolas Toussaint, Lindsay Munroe, Suryava Bhattacharya, Gina Sajith, Ei Lin, Eeshar Singh, Ka Yee Kelly Chu, Saleha Kabir, Kuberan Pushparajah, John M. Simpson, Julia A. Schnabel, Alberto Gomez

**Affiliations:** 1School of Biomedical Engineering & Imaging Sciences, King’s College London, London SE1 7EU, UK; shujie.deng@kcl.ac.uk (S.D.); gavin.wheeler@kcl.ac.uk (G.W.); nicolas.toussaint@gmail.com (N.T.); lindsay.munroe@kcl.ac.uk (L.M.); suryava.bhattacharya@kcl.ac.uk (S.B.); gina.sajith@kcl.ac.uk (G.S.); ei.lin@kcl.ac.uk (E.L.); singhsantoor@gmail.com (E.S.); ka.y.chu@kcl.ac.uk (K.Y.K.C.); kuberan.pushparajah@kcl.ac.uk (K.P.); john.simpson@gstt.nhs.uk (J.M.S.); julia.schnabel@kcl.ac.uk (J.A.S.); 2Department of Congenital Heart Disease, Evelina London Children’s Hospital, Guy’s and St Thomas’ National Health Service Foundation Trust, London SE1 7EH, UK; saleha.kabir@gstt.nhs.uk; 3Department of Informatics, Technische Universität München, 85748 Garching, Germany; 4Helmholtz Zentrum München—German Research Center for Environmental Health, 85764 Neuherberg, Germany

**Keywords:** virtual reality, pre-operative imaging, echocardiography

## Abstract

The intricate nature of congenital heart disease requires understanding of the complex, patient-specific three-dimensional dynamic anatomy of the heart, from imaging data such as three-dimensional echocardiography for successful outcomes from surgical and interventional procedures. Conventional clinical systems use flat screens, and therefore, display remains two-dimensional, which undermines the full understanding of the three-dimensional dynamic data. Additionally, the control of three-dimensional visualisation with two-dimensional tools is often difficult, so used only by imaging specialists. In this paper, we describe a virtual reality system for immersive surgery planning using dynamic three-dimensional echocardiography, which enables fast prototyping for visualisation such as volume rendering, multiplanar reformatting, flow visualisation and advanced interaction such as three-dimensional cropping, windowing, measurement, haptic feedback, automatic image orientation and multiuser interactions. The available features were evaluated by imaging and nonimaging clinicians, showing that the virtual reality system can help improve the understanding and communication of three-dimensional echocardiography imaging and potentially benefit congenital heart disease treatment.

## 1. Introduction

Congenital heart disease (CHD) affects nearly one in every 100 liveborn infants. In children under five, the leading causes of death in the developed world are premature birth and congenital anomalies, each accounting for 25% of the total. Moreover, the most frequent cause of death in the congenital anomaly group is CHD, which can present in a vast number of different forms and with major individual variation between patients, which is critical to understand for care planning. Thus, accurate imaging prior to procedures is vital. Decisions relating to the type and timing of cardiac procedures are made at multidisciplinary meetings following the review of standard imaging modalities including echocardiography (echo). Improvements in imaging data prior to surgery has reduced morbidity and mortality, which not only results in better care, but also reduces cost.

Essentially, the complex, three-dimensional (3D) and dynamic nature of CHD makes interpretation, quantification and communication of patient-specific aspects of the disease, both between clinicians and with patients and their families, extremely challenging. Recently, 3D-printed models of the heart have been produced to plan procedures in complex disease. For example, a recent multicentre study showed that this impacted the surgical approach in 44% of complex cases [1], including cases where surgery was not thought to be feasible at all. Three-dimensional printing has also been used to communicate CHD to patients and members of the public and has proven to be an effective tool [2]. However, 3D printing is time consuming (both for generating the surface models and for printing them) and expensive and can only represent static anatomy. Immersive computational techniques, such as Virtual Reality (VR), may allow for improved visualisation and understanding of cardiac anatomy in CHD [3].

In this paper, we present a VR system for improved image-based planning of complex cardiac procedures, with application to CHD. Specific features of the system have been individually published before, including visualisation, measurement and interaction capabilities and evaluation in a clinically realistic scenario. In this paper, we compile this previous work and present it as an integrated system.

### 1.1. Related Work

Immersive Extended Reality (XR) technology, which includes Augmented Reality (AR), Virtual Reality and Mixed Reality (MR), has been proposed [4,5,6,7,8,9,10,11,12,13,14,15] to improve the visualisation of, and interaction with, anatomical models of the heart for educational and medical applications.

Most educational systems use cardiac models that have been hand-crafted [4] to communicate a realistic looking, yet idealised from a structure perspective, model of the heart, which can often be cropped to look at each part independently using VR [4,5] or as a mobile AR app [6,7]. These systems are targeted at teaching unspecific, idealised cardiac anatomy. Improving on these, patient-specific models have been built and used in XR devices to support surgery and interventions [8]. For example, surface mesh models obtained by manual segmentation of computed tomography (CT) angiography or cardiac magnetic resonance (CMR) images have been used for cardiac surgery planning in CHD [9] and to support interventionists during procedures using mixed reality headsets [10]. The reason why surface models have been used since the first medical XR applications is that meshes are easier and faster to render and inspect (e.g., slice, clip or crop) and represent deterministic shapes that can be measured without ambiguity. However, computing such models is a challenging and time-consuming task. Despite the active research in automatic segmentation of cardiac images [16], accurate segmentations still require a substantial amount of manual interaction [17]. This is especially true in images of CHD patients due to the particularly large interpatient variability, limiting the accuracy of statistical population-based methods. Moreover, segmentation is particularly challenging in echocardiography, which is the modality of choice in all initial cardiac examinations and a key modality in CHD surgery and intervention planning [18], particularly any procedure involving valves and looking at motion where echocardiography is superior, and complementary, to CMR [19] and CT.

Our proposed system removes the need for segmenting cardiac structures and instead uses the raw echocardiography image data directly in a VR system, by using volume-rendering techniques. As described in more depth in Section 2.2, this is achieved by combining the VR development framework Unity https://www.unity.com (accessed on 30 May 2021) with a medical image visualisation library, VTK https://www.vtk.org (accessed on 30 May 2021). In recent years, other existing medical imaging software solutions have also incorporated VR technology into their existing medical imaging and volume-rendering tools. MeVisLab https://www.mevislab.de (accessed on 30 May 2021) integrated a VR headset to enable medical applications using OpenVR [11], for a proof-of-concept of VR visualisation of 3D CT images. 3D Slicer VR [12], an extension that enabled communication between 3D Slicer https://www.slicer.org (accessed on 30 May 2021) and a VR headset, was introduced, effectively enabling VR visualisation of rendered echocardiograms. The latest developments of SlicerVR [13] allow sharing of a session between two users over the network and extend some of the existing Slicer features to VR. These solutions are aimed at enabling VR visualisation and interaction in existing general-purpose imaging platforms and, as a result, have not been developed with the purpose of assisting in cardiac procedure planning and, therefore, lack some desirable features, such as measurements, annotation and simultaneous or deferred sharing and collaboration among users.

Although the advent of new headset and display technology and improvements in graphics and computing are enabling alternative XR modalities, such as AR and MR [20,21], VR still has advantages over these for developing, evaluating and deploying an immersive procedure-planning system. In VR, the entire scene is virtually created, hence providing full control of what to show to the operator. In mixed and augmented reality, the real scene needs to be accurately mapped to enable the integration of virtual and real objects and augmented information. Such integration comes at a computational and economic cost and might also introduce errors, potentially leading to incorrect image interpretation. Additionally, VR allows disengagement from the real physical location of the operator, lending itself naturally to remote collaboration and communication where physical collocation is not possible. In summary, VR is an ideal choice of immersive technology to investigate a fully featured system for cardiac surgery and intervention planning.

A number of commercial VR systems that allow immersive visualisation of cardiac images are available, and for completeness, we include the most relevant here. Echopixel https://www.echopixeltech.com (accessed on 30 May 2021) has demonstrated their stereoscopic-type display for immersive visualisation of echocardiographic CHD data, which, thanks to a stylus interaction tool, allows for accurate measurements [14]. Medical Holodeck https://www.medicalholodeck.com (accessed on 30 May 2021) provides a general-purpose VR app that can be used for immersive interrogation of 3D medical images. Medical Holodeck and Bosc (Pyrus Medical systems) have been applied to VR-based surgery planning with positive results [15].

In summary, there is a clear academic and industry trend towards immersive visualisation of medical images for improved understanding of complex anatomy and strong evidence that clinicians are favourable to adopting it. Virtual reality technology is ideally placed to support interaction and interrogation of 3D echo intuitively and with full perception of depth.

### 1.2. Contributions and Paper Organisation

In this paper, we present a complete VR system for cardiac procedure planning in CHD patients. Our contributions are:We review the visualisation, quantification and interaction features of our system that have been individually published elsewhere and present them in the context of a complete system; we propose new experimental features that are currently under development;We present a comprehensive assessment of the system, by compiling evaluations carried out on the individual features and on the system as a whole.

The paper is organised as follows. Section 2 first describes the hardware (Section 2.1) and then the software. A first overview of the software architecture is provided in Section 2.2, followed by a description of features related to visualisation (Section 2.2.2), interaction (Section 2.2.3) and multiuser utilisation (Section 2.3). The experiments and evaluation sessions are included in Section 3. For the sake of clarity, and given the large number of different evaluations, we show the results of each evaluation together with its description.

## 2. Materials and Methods

In this section, we describe our VR system for improved image-based planning of complex cardiac procedures. The system includes a hardware and a software component. The hardware consists of a VR headset and a high-end workstation. The software component renders 3D echo images and provides interaction and measurement tools and other advanced features. Both are described in more detail below.

### 2.1. Hardware

For the VR equipment, we chose the HTC Vive https://www.vive.com (accessed on 30 May 2021) over competing hardware including Oculus Rift https://www.oculus.com/rift (accessed on 30 May 2021) and non-VR options such as Microsoft Hololens https://www.microsoft.com/en-us/hololens (accessed on 30 May 2021) and Meta 2 https://www.metavision.com (accessed on 30 May 2021). The HTC Vive offers a number of advantages: high-resolution display and high-quality tracking for a natural immersive experience; wide compatibility with VR software frameworks such as Unity; intuitive and easy-to-use controllers; and tethered computing, allowing the upgrade of computational resources for more demanding features without the need for a different headset. In addition, the HTC Vive is widely used and tested, is sterilisable using UV light, enabling clinical use, and is available at an affordable price that can facilitate adoption in the routine hospital setting.

The HTC Vive needs to be connected to a computer equipped with a high-end graphics processing unit (GPU) to carry out the stereoscopic rendering at high frame rates. All experiments were carried out with a Dell Alienware laptop with an Intel i9-8950HK 2.90GHz CPU, 16GB RAM, Nvidia GTX1080 GPU with 8GB RAM or a Dell Alienware desktop with an Intel i7-8700 3.20GHz CPU, 32GB RAM, Nvidia GTX 1080Ti GPU with 11GB RAM.

### 2.2. Software: Core Features

We built a Unity application that leverages the native plug-in system to extend Unity’s features for medical imaging. Unity is a cross-platform environment for developing 2D, 3D, VR and AR video games on many mobile, desktop and web platforms. Unity supports some of the most popular XR APIs and hardware, such as Oculus XR, Windows XR and OpenVR, and as a result, was a convenient choice to develop our technology. An illustration of the graphical interface during a planning session is shown in Figure 1. An overview of the application and the core features for image visualisation and interaction is described in the following sections.

#### 2.2.1. Integrating Medical Imaging and Virtual Reality

Unity’s features can be efficiently extended via a low-level native plug-in interface. We used this interface to incorporate medical imaging capabilities via VTK. VTK is an open-source C++ library aimed particularly at medical imaging visualisation. Crucially, VTK implements OpenGL rendering, which can be used in external applications. This enables OpenGL context sharing between VTK and Unity using Unity’s native plug-in interface.

For our application, we developed a VTK-based plug-in for Unity [22] that enabled VTK rendering in Unity in VR. The plug-in has been made available open-source https://gitlab.com/3dheart_public/vtktounity (accessed on 30 May 2021). Our application uses VTK to load medical images and to render them both as volumes and as MPR slices. All other components of the application were built using Unity assets.

#### 2.2.2. Visualisation

Volume render display is important to accessible VR display, enabling use by nonimaging clinicians—since it most closely resembles the anatomy. While volume rendering is the focus of accessible imaging, display of 2D slices through the volume as Multiplanar Reconstruction (MPR) images is also required by clinicians. Imaging specialists are familiar with MPR images, and so, we hypothesised that their presence would provide confidence to these users. Additionally, many current measurements are based on 2D slices, and as a result, being able to recreate and display 2D MPRs aids confidence in and hence the adoption of the technology.

#### 2.2.3. Interaction

A major reason for the accessibility and popularity of 3D-printed models is that they may be freely picked up, rotated and moved without conscious effort. Conversely, these same operations are often complex for nonexpert users in 2D medical visualisation applications. VR offers control whose intuitiveness approaches that of 3D prints.

VR handheld controllers are designed to be easy to learn and use, allowing users to adopt them quickly. Additionally, to maintain accessibility, we minimised the buttons used and the overlay function icons on the buttons. Overall, users typically find handheld controllers to be simpler to use than using 2D controls and a screen because the controllers provide interactivity in 6 degrees of freedom (DoFs), which is more intuitive to interact within a 3D world than the 2 DoF provided by conventional 2D interaction methods such as a mouse.

User interaction is implemented entirely in Unity using its physics system. Interactable objects within the scene (for instance, the volume render, a zone just in front of the “nose” of the controller and a number of 3D interactive gizmos described below) are given physics collider components, for which C# scripts are used to manage interactions between them and to track the object currently selected by the user. User feedback is given by a haptic buzz upon the entry and exit of any interactable object, which is also highlighted (e.g., brighten its colour or adding a halo depending on the object). Objects are picked up by pressing the controller’s trigger.

Figure 2 illustrates the interactable objects and gizmos that enable user interrogation of the data. The core interactions implemented in our system, and the means to use them, are:Pick up and rotate the volume: We visualise the volume within a bounding box, which is interactable, for picking up, then moving and rotating the volume (Figure 2a);Scaling: Each corner of the bounding box is designed for scaling the volume (Figure 2a);Cropping plane: In a scan, the anatomy of interest is often located in the middle of the volume, obscured by the surrounding tissue. Cutting into the volume using a cropping plane is the most common tool to access and visualise the anatomy of interest. Direct placement of the plane in 3D using the controller makes this easy for the user (Figure 2b);Windowing: The image brightness, or gain, and contrast can be interactively adjusted by the user. Two parameters, window width and window level, are set using the controller’s touchpad, and the volume rendering is updated in real time;Animation: For dynamic 3D echo, animation is implemented by looping through a sequence of volumes at a preset frame rate. There are user controls for play and pause and to step forward and backward frame by frame. This is controlled using a virtual 3D panel attached to a controller (Figure 3b);Landmark placement: Navigation in CHD 3D echo images can be complex due to the cropping required to access the anatomy of interest, the lack of orientation indicators that exist in CT and MR and the complexity of the anatomy. A predetermined set of user placeable landmarks addresses this. These each have a label and a point, the two connected by a line. The user can pick up and place the label and the point independently or place the whole marker as one (Figure 2c);Measurements: Quantification of medical images is essential for treatment decisions. The simplest and most commonly used measurement is line measurement—the distance between two points. VR allows the user to make line measurements by placing the start and end points directly on the volume, without being restricted to a 2D plane (as is common in 2D applications).The measurement widget was implemented entirely in Unity and contains the start and end points, a label with the distance and connecting lines (Figure 2d). The points and label may be picked up individually, or the entire widget may be picked as a whole, and moved by the user. To allow the user to “zoom in” to examine a small area of the volume, the measurement’s widget size is kept constant as the volume’s scale changes.

### 2.3. Software: Multiuser Features

The multiuser feature aims at enabling collaboration, education and general communication purposes, such as consultation between clinicians, explanation between clinicians and patients or demonstration between teacher and students. We considered two main use cases: first, simultaneous (and synchronous) usage by multiple users, for example for real-time scenarios such as clinician–patient communication, for which the ability to share the virtual scene is required; second, scenarios such as a case review to provide a second opinion, where the ability to record a case investigation and play it back asynchronously at an arbitrary time is required. These two features are described in more detail below.

#### 2.3.1. Synchronous Sharing

This feature enables real-time or online visualisation and interaction in the same virtual scene between users on either VR or 2D visualisation platforms via network connections.

This function is implemented with the client–server architecture based on cloud technology, using the Photon Unity Networking (PUN) asset https://www.photonengine.com/pun (accessed on 30 May 2021). Multiple users as clients can connect to the server on the cloud. Only one user, i.e., the main user, can interact with the scene to avoid conflict, and the resulting scene updates from the interactions will be propagated to all others. Roles can be swapped upon request. When viewed on another user’s platform, each individual user’s headset and controllers are rendered as a virtual avatar (Figure 3a).

To optimise the sharing experience, large data objects (mainly imaging data) are loaded and visualised from a local file system on each user’s device. As a result, only a few parameters of all these objects will be transmitted (position, orientation of objects and visualisation settings), and only when they are updated.

#### 2.3.2. Asynchronous Sharing

This feature enables offline, or delayed, access of an existing session using recording and playback mechanisms. A session is an arbitrary collection of interactions described in Section 2.2.3, and all these can be tracked and stored using the offline sharing feature.

This feature is built on a time-ordered list of time-stamped “keyframes”, each of which stores the parameters that define the instantaneous state of the scene. In recording, a list of such keyframes is saved chronologically in a file. In playback, the file is loaded, and the instantaneous scene states are replicated in order.

The list can record keyframes at regular intervals, or every time the scene changes. In addition, the user can bookmark relevant “keyframes” for random access in playback. Playback is implemented based on timestamp retrieval. If the input timestamp lies between two adjacent keyframes, the system outputs a linearly interpolated state between the two.

### 2.4. Software: Experimental Features

In this section, we describe three additional features (illustrated in Figure 4) that could help improve usability, consistency and efficiency, but may not be essential for a clinically useful system, and have been implemented, but not yet tested in a clinical evaluation. Preliminary performance results are briefed here.

#### 2.4.1. Blood Flow Visualisation

The blood flow pumping efficiency of the heart is often affected by CHD. Surgery and interventions aim to restore this to the greatest extent possible. Computational Fluid Dynamics (CFD) simulation enables the flow of blood in the heart to be computed before being visualised and analysed [23], and so aids treatment planning. CFD data are fundamentally 3D and often complex, so VR could provide an aid in their understanding by improving visualisation and interaction.

We extended our Unity plug-in to take advantage of VTK’s CFD rendering capabilities to prototype CFD visualisation in VR [24]. A CFD dataset modelling the right ventricle of a patient with Hypoplastic Left Heart Syndrome (HLHS) was loaded into VTK. The dataset included anatomy (as a triangulated mesh), blood pressure (shown on a user-defined plane) and blood velocity (represented with streamlines).Our Unity-VTK integration allowed display and user control of the positioning of the mesh, pressure plane and streamlines, as shown in Figure 4a.

#### 2.4.2. Haptics

Haptic feedback can aid interaction data and increase immersion in a VR scene [25]. We investigated a simple form of haptic feedback, using the vibration function in the HTC Vive controller and a pregenerated mesh representing the interaction surface [26].

A preliminary usability study was set up, where 10 nonclinical participants carried out measurements on synthetic data with and without haptic feedback enabled. The results showed no significant improvement in measurement accuracy [26]. However, 90% of the participants felt that the haptic cue was helpful for deciding where to place the measurement point, and 88.9% of the participants felt more immersed in the VR scene with haptic feedback (Figure 4b). Overall, 70% of users expressed a preference for the haptic system.

#### 2.4.3. Automated Anatomic Orientation

Users often need to orient a 3D echo image by manually interrogating it. This is made challenging by the limited field of view, obstructing structures, patient-specific structural abnormalities and the lack of anatomical orientation markers. We developed a method for automatic orientation of the rendered volume using a deep neural network to estimate the calibration required to bring the 3D image to anatomical orientation [27,28]. This calibration is applied to the 3D image aligning with a reference anatomical model shown next to it. This anatomical model tracks any rotations the user makes on the 3D image, thus serving as an orientation cue (Figure 4c). Our preliminary results showed re-orientations to an average Mean Absolute Angle Error (MAAE) of $9.0^{\circ }$.

This method was implemented in Python and integrated into Unity using socket communication implemented in C# scripts. Our VR application re-orients the echo image upon receiving the predicted orientation, so that it matches the reference model.

## 3. Experiments and Results

We carried out three clinical evaluation sessions, in which clinicians of different specialisations within the area of cardiology used the system with images from CHD patients. The evaluations were focused on testing the features described in Section 2 in terms of clinical usability, measurements and the clinical benefit of using our system.

### 3.1. Clinical Usability and Immersive Experience

This evaluation was focused on the clinical usability and acceptability of the software in terms of visualisation [29] and interaction [30] elaborated in the previous Section 2.2.2 and Section 2.2.3:**Benchmark:** QLAB 10.8 software (Philips), which is widely used in clinical practice;**Data:** A 3D+time transesophageal echo image from one patient with a morphologically normal heart. Institutional ethical approval and patient and participant consent were acquired for the study;**Participants:** The evaluation was conducted in the Evelina London Children’s Hospital, from which 13 clinicians with various experience levels (1 trainee, 4 junior and 8 senior) volunteered to participate, comprising 5 imaging cardiologists, 5 cardiac physiologists, 2 cardiac surgeons and 1 cardiac interventionist. The majority (9) of them were familiar with QLAB and used it at least weekly. Just over half (7) had never used VR, and the remaining 6 had used it twice or less;**Procedure:** A VR training session was conducted for each participant, until they were confident about the essential interactions, including: picking up, moving and placing scene objects (e.g., the volume, a landmark, the cropping plane), adjusting gain and contrast and controlling animation playback. Following that, each participant evaluated (1) image quality and (2) interaction. A questionnaire was completed after all evaluations were performed. Each trial took approximately 30 min.

When evaluating the image quality, each participant was presented with the QLAB image first, then the VR image, both with the same preset view, default gain/contrast and default playback speed (Figure 5). The participants were asked to examine the rendering quality and depth perception. Interactions were not allowed in this stage, but in VR, the participants could freely move their head when evaluating the depth perception.

When evaluating the interactions, the participants were asked to use all the interactions learnt in the training session following the experimenter’s instructions, and then to freely explore the VR scene.

**Result:**Table 1 shows a clear preference for the VR system in terms of image quality. A one-sample Chi-squared test was applied to each row of the result. The significance level was set to 0.05. Note that the degrees of freedom were 2 for the preference for Colourmap and Overall and 1 for preference for Resolution and Depth because one column of their results was 0, which was ignored in the significance tests. The participants significantly preferred the Resolution and Depth of VR, as well as the Overall preference. The difference of their preference for Colourmap was insignificant. In addition, 12 out of 13 clinicians agreed that the image quality was adequate for clinical use.

The comfortableness was evaluated regarding the side effects of using the VR system, such as dizziness, nausea, headache, sore neck, etc. One participant found the proposed VR system “Somewhat Uncomfortable”; the remaining 12 participants found the VR system comfortable. However, three participants identified uncomfortable aspects of the VR system, including minor vision abnormalities, difficulty fitting their spectacles into the headset and mild dizziness afterwards.

In terms of the learning curve, most users found it easy to learn and use for all basic interactions introduced in Section 2.2.3 (Table 2). When asked which system would be preferred, given the same features in both, 11 indicated preference for the VR system, and 2 indicated equal preference.

Overall, our proposed VR technology was perceived by clinicians as easy and acceptable for clinical use, capable of appropriate image quality and providing a good, intuitive user experience.

### 3.2. Clinical Measurements

This evaluation focused on assessing our VR line measurement tool against two commercial 3D echo visualisation and analysis tools, in terms of accuracy and precision [31,32]:**Benchmark:** QLAB 10.8 (Philips) and Tomtec CardioView, both widely used commercial tools that allow 2D (QLAB) and 3D (Tomtec) measurements;**Data:** Echo images obtained on a calibration phantom <add model> and on 4 paediatric patients with CHD;**Participants:** 5 cardiologists, comprising 3 imaging cardiologists and 2 physiologists, of whom 4 were senior (5+ years of experience) with 1 junior (<5 years experience). All participants used QLAB almost daily and Tomtec at least once a month. VR use for 3 participants was “nearly every month” and the remaining 2 only “rarely”;**Procedure:** Measurements were made on MPR in QLAB, on volume render in Tomtec and in volume render in our VR application. Participants made 6 measurements on the echo image of the phantom, and 5 clinically meaningful measurements at specified frames on each of the patient echo clips. Before recording evaluation measurements on our VR system, participants were free to practice making measurements. For all applications, participants were asked to prioritise the accuracy of measurement and were free to explore the data, alter gain and contrast, and so on. In all cases, the measurement distance was hidden from the user. After making the measurements, each participant completed a questionnaire;**Results:** Measurements on the phantom have a known true value. Comparison to these showed that VR measurements were the most accurate of the 3 tools for the 2 smallest cylinders, and QLAB the most accurate for the largest cylinder. In all but one case, QLAB was the tool with the smallest measurement variability.

Bland–Altman plots were created to compare Tomtec and VR against QLAB for patient data measurements (Figure 6). The plots show that, compared to QLAB, VR measurements had a lower bias than Tomtec ones, but larger variation.

Participants were most confident in their measurements made in QLAB, which is the most frequently used application for this task, and where they identified the use of (orthogonal) MPR plane images as contributing to their confidence. Confidence in VR measurements was lowest of the three tools. In addition to being a new technology, contributing factors were the blurriness of the render and the impact of gain. However, the ability to gain an overview of the anatomy and the intuitiveness of use were positives.

### 3.3. Clinical Benefit

This evaluation focused on assessing whether the proposed system was of clinical benefit in planning surgery for CHD patients [33]:**Benchmark:** Philips QLAB 10.8 (Philips);**Data:** Retrospective echo data from 15 children requiring surgery on the AV valves were collected. At the time of the experiment, all patients had undergone surgery;**Participants:** A group of 5 paediatric cardiothoracic surgeons with various experience level (3 with >15 years experience, 1 senior trainee >3 years, 1 junior trainee <3 years), who were not involved in the surgeries and blinded to their outcome, participated in this evaluation. Three of them used VR once or twice, and the others never used VR. A tutorial and practice was given for them to familiarise themselves with the system;**Procedure:** The surgeons evaluated 3 cases each. For each case, they were presented with the clinical details of the case including preoperative images on a standard system, analogous to our normal clinical practice, completed a questionnaire, then presented with the images on our system and completed a questionnaire again. We investigated if using our system made the surgeons change their initial surgery plan; and whether those changes matched findings during surgery;**Results:** After using our system, the surgeons experienced an increase in their confidence regarding the surgical approach in two thirds of the cases and reported that they would have modified their original plan in 60% of the cases. Our experiment provided evidence that our system can improve surgery planning by facilitating the understanding of complex anatomy for surgeons before the procedure. A more detailed description of the study and a discussion of the results has been published [33].

## 4. Discussion

Our results showed that the proposed VR system is clinically acceptable, easy to use and accurate and has the potential to improve surgery planning in CHD.

The VR system’s features were well accepted by clinical users, both in terms of usability and quantification. Regarding usability, the vast majority of participants found the system comfortable to use. The minor discomfort feedback received is likely to be addressed by manufacturers, who improve headsets to make them more ergonomic in each iteration. Good image quality, intuitive interactions, ease of use and learning and improved three-dimensional perception of the anatomy were described by the participants as clear benefits over standard 2D systems. Overall, the participating clinicians, and particularly surgeons, were positively engaged with the technology and the prospect of using it in their practice. In terms of quantification, validated tools are essential for clinical uptake of VR. As a crucial first step, we showed that line measurements made in a volume render in VR showed no statistically significant bias compared to Philips QLAB, the most frequently used 2D echo measurement software at our institution. However, the wider variation in measurements compared to Tomtec’s CardioView deserves further investigation. Potential sources of this increased variability arising from the application include the gain and contrast settings chosen for the volume rendering and the depth precision of the point placement. Linking the measurements to the MPR images from the volume render may address this variability and increase confidence.

Clinical adoption of any new technology is a significant challenge. For example, we anticipated that some users may find the VR experience isolating or the controllers difficult to use. For this reason, we put substantial effort into (1) making the system easy to use, (2) making its continued use comfortable and (3) initially evaluating features separately to gain clear feedback. This contributed to the excellent attitude of clinicians towards our system, but also poses interesting challenges for the future: (1) ensuring that ease of use is maintained as we incorporate new features; (2) maintaining high frame rates with advanced visualisation techniques, more sophisticated rendering and new hardware that supports higher resolutions; (3) carrying out prospective evaluations of the technology.

We presented a selection of our experimental features that are being developed and tested with these aspects in consideration, specifically flow imaging, assisted image orientation and haptic feedback. An interesting outcome when testing the latter was that, while haptics may only marginally improve measurement accuracy, it facilitated a more immersive user experience, improving confidence and increasing the ease of use. However, this technique shares some compromises with 3D printing—namely that boundaries can change depending on the gain and contrast chosen and obtaining a high-quality segmentation often requires time-consuming and skilled manual intervention.

We showed that our VR system provided surgeons with a better pre-operative understanding of the anatomy, facilitating better surgical planning choices. This was only evaluated with retrospective cases, and future work will include prospective evaluations. Furthermore, more investigations into the challenges for adoption and technology deployment in a hospital environment should be carried out, one such question being the space required. Many VR systems require a dedicated room with a large amount of free space for the user. However, in many hospitals, this is a difficult requirement to meet. Our VR system started from the principles of 3D-printed models, showing data close to life size, and everything is within arm’s reach. Thus, the space requirements are relatively small. Indeed, users often sat down to use the system, and we ran many evaluations at regular office desks. This smaller-scale version of VR aids integration into hospital life.

From our experiences, we learned that using nonmedical VR apps, such as games, can be useful to show potential users what the possibilities of the technology are, and they often related those possibilities to clinical needs and feature requests for our system.

## 5. Conclusions

We presented a virtual reality system for procedure planning using echo images for surgery and interventions of congenital heart disease patients. The usability of the system and the core features were evaluated by clinicians, separately and as a whole.

Our system is clinically acceptable, easy to use, intuitive to learn and accurate. Additionally, evidence suggests that it may improve procedure planning through a better understanding of anatomy.

## Figures and Tables

**Figure 1 jimaging-07-00151-f001:**
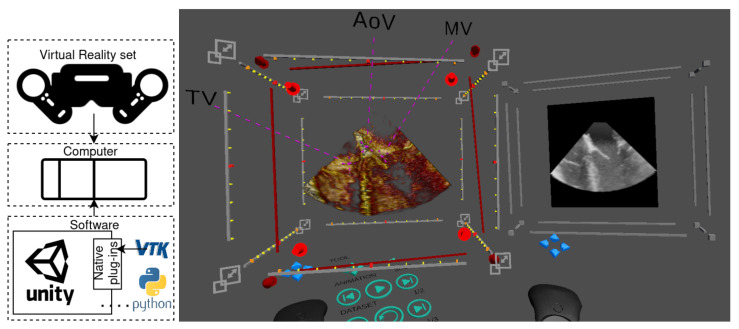
System overview (**left**) including the hardware (VR set and computer) and software (Unity application connected to Python and to VTK through Unity’s native plug-ins) and an example of a screen capture of our virtual reality application (**right**). This case shows a cut of a 3D ultrasound image of the left ventricle with three highlighted landmarks (TV—tricuspid valve, AoV—aortic valve, MV—mitral valve) and a multiplanar reconstruction plane showing the cropping slice.

**Figure 2 jimaging-07-00151-f002:**
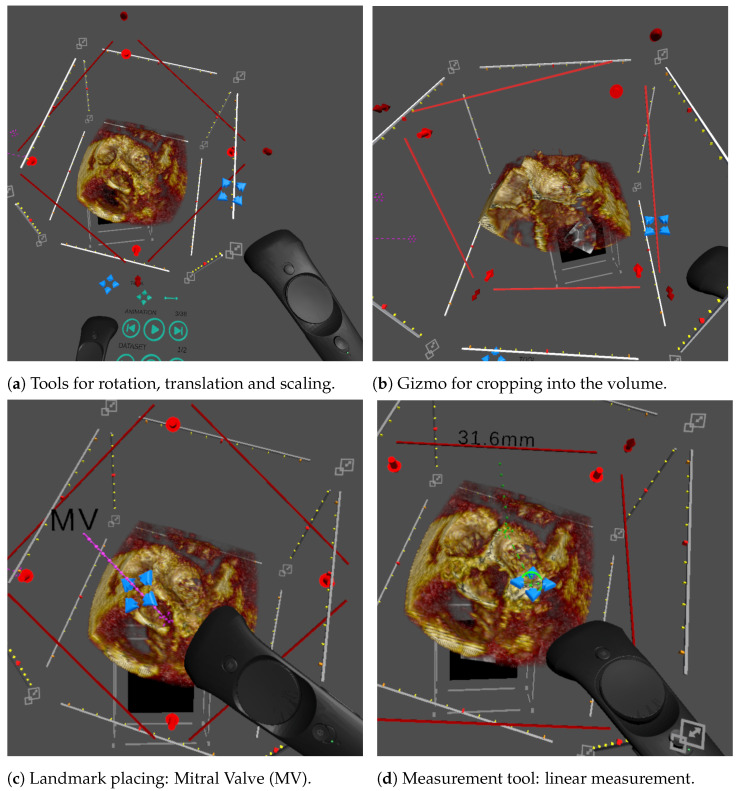
Illustration of the different interactions featured in our VR system, controlled using the blue cross-hair widget attached to the controllers. (**a**) Rotations and translations are highlighted using the volume bounding box. Scaling is applied by pulling the corners of the bounding box. (**b**) Cropping uses a transparent red plane that cuts into the volume. (**c**) Landmarks include a pin point, a label line and a label and can be moved and stretched with the cross-hair tool. (**d**) The linear measurement tool allows the user to draw a green dashed line between two end points and displays the length in mm.

**Figure 3 jimaging-07-00151-f003:**
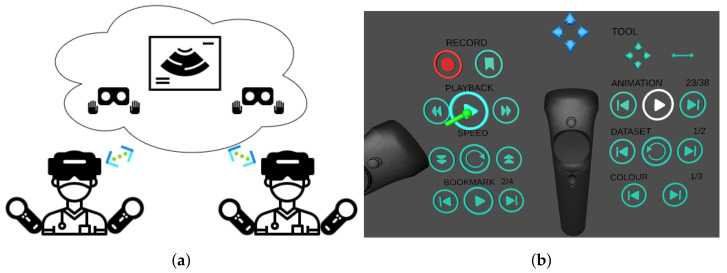
Multiuser features. (**a**) Representation of synchronous collaborative planning, where two users simultaneously interrogate the data remotely. Each user sees the other users by their virtual avatar. (**b**) Interface for asynchronous collaboration, where one user can play back an interrogation session previously carried out by another user. The playback is a single-player feature who can toggle the display of the avatar who performed the recorded interaction.

**Figure 4 jimaging-07-00151-f004:**
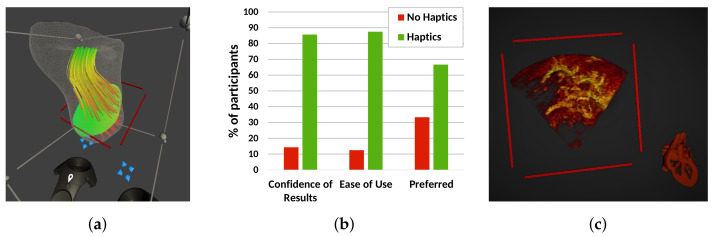
Experimental features investigated using our VR system. (**a**) Visualisation of a CFD blood flow simulation in the right ventricle of an HLHS patient, showing blood velocity with streamlines and a user-defined plane with the pressure distribution over that plane. (**b**) User opinion on using haptic feedback during a measurement task, showing overall preference for enabled haptics. (**c**) Automatic anatomical orientation of the rendered volume that is aligned with a cardiac model.

**Figure 5 jimaging-07-00151-f005:**
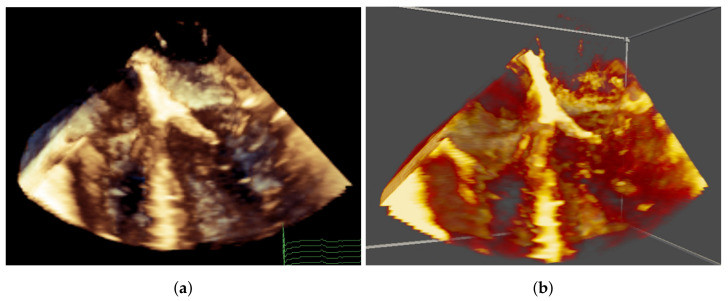
Comparison of image quality between QLAB and the VR system. (**a**) QLAB image with colour depth cueing, where closer tissues are brown and farther tissue are blue. (**b**) VR image with a similar colour map, but without colour depth cueing.

**Figure 6 jimaging-07-00151-f006:**
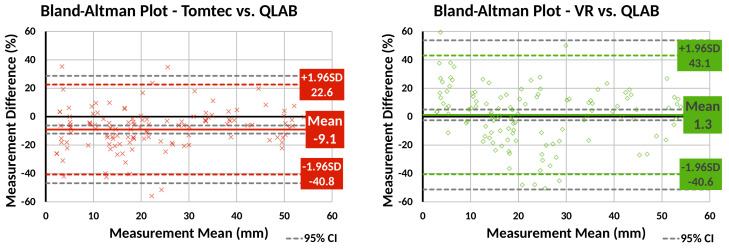
Bland–Altman plots comparing line measurements made using our VR system and Tomtec CardioView to Philips QLAB. Our VR system’s measurements show less bias than Tomtec, but greater variability.

**Table 1 jimaging-07-00151-t001:** Preference comparison of image quality between the proposed VR system and QLAB. A $\chi^{2}$ test shows that participants significantly prefer the VR system overall, and also specifically in terms of depth and resolution.

	QLAB	Same	VR	$\chi^{2}$	*p*-Value
Colourmap	3	2	8	4.769	0.092
Resolution	2	0	11	6.231	**0.013**
Depth	1	0	12	9.308	**0.002**
Overall	1	1	11	15.385	**0.000**

**Table 2 jimaging-07-00151-t002:** User assessment of essential interactions of the VR system in terms of ease of learning and ease of use. All interactions were considered easy or very easy by the majority of participants, with none finding them very difficult.

		Very Difficult	Somewhat Difficult	Easy	Very Easy
Learning	Volume	0	1	4	8
	Cropping	0	0	6	7
	Landmarks	0	0	6	7
	Windowing	0	1	7	5
	Overall	0	1	5	7
Use	Volume	0	1	3	9
	Cropping	0	0	5	8
	Landmarks	0	1	5	7
	Windowing	0	2	6	5
	Overall	0	0	6	7

## Data Availability

The data presented in this study are not publicly available due to restrictions in the research ethics.

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
