# Peer review of "A Virtual Reality System for Improved Image-Based Planning of Complex Cardiac Procedures"

_2313-433X, 2021, doi:10.3390/jimaging7080151_

Round 1
Reviewer 1 Report
General comments:
The paper has an interesting topic and describes an innovative solution for 3D echocardiographic holographic visualization that deserves publication. My review will be clinically oriented and less focused on the technical choices per se.
As stated, part of the data that is published already. Thus the paper is a combined review of own work, with an addition of a clinically oriented user oriented evalutation of the solution. This combo of review and original data should be clearly expressed already in the first introduction part, eg line 40 p 2, so that the reader at an early stage is made aware of what is old and what is new content. The aim of the paper should thus become even clearer for the reader.
The manuscript seems a bit long and “word-rich” and deserves another round of tightening up. Pathos-filled words like “overwhelming” may be bit overwhelming and should be avoided. ( P11 line 379 : suggest dropping words like overwhelming. Rather use clear or highly significant. )
Specific comments:
Abstract: I suggest adding “potential” before benefit in line 14 - as a true benefit in treatment, in terms of improved outcome for the patient, was beyond the scope of this paper.
1.1 line 57 : The paper cited here (reference no 10) does not describe the intraoperative use of hololens by heart surgeons - as this would be surprising due to physical conflict between the hololens and the surgeons magnifying glasses. It describes EP work – so skip the surgeons here.
2.4 The three presented future developments of the application: blood flow, haptics and auto -orientation, are all exciting features, but being at a POC Level and partly previously published they should be presented later in the paper and be given less space.
3.2. The very low precision of VR vs QLAB in patients should be discussed more balanced – underpinning its limitations for clinical use. Accuracy without precision is not useful at an individual level. A linear measurement tool used in volume rendering of echo has some serious challenges to be solved.
- Discussion:
Again – modify the statement can improve surgery planning. Line 459. We don’t know so far that it can improve surgical planning.
Elaborate a bit more on the problems related to precision of measurements in VR.
Institutional Review Board Statement : missing.
Supplements:
Suggestion: questionnaires should be published as supplementary files. Reading the paper I repeatedly wonder about how the statements presented in the user study are formulated in terms of producing unbiased answers. Videos of holographic volume rendering vs QLAB rendering could be interesting as supplements as well.
In summary: Good technical work. Deserves a compiled publication like this – but suggest toning down a bit the highly positive description and balance it with some elaboration on the challenges, primarily regarding measurements.
Reviewer 2 Report
The authors demonstrate a virtual reality system for planning cardiac surgery. The manuscript is well written with adequate figures, and the results show the potential of the system to be used in the fields. I think the manuscript is acceptable after some minor corrections.
- The schematic illustration of the configuration or image processing procedure of the system would be helpful to understand the overall system.
- Some words in the Figures are not clearly visible. ‘TV’, ‘AoV’, ‘MV’ in Figure 1, for example. I would change the color to white since the background color is dark.
- In the captions of figures, the author should re-address the abbreviations. The reader should understand without finding abbreviations in the other part of the article.
- Figure 3, I would annotate (a) and (b) rather than ‘left’ and ‘right’.
- In Figure 6, did you averaged the signed difference? Not the absolute difference? It would be reasonable to analyze with the absolute values of the differences.
